# Cross-Cultural Adaptation, Reliability, and Validity of a Hebrew Version of the Physiotherapist Self-Efficacy Questionnaire Adjusted to Low Back Pain Treatment

**DOI:** 10.3390/healthcare11010085

**Published:** 2022-12-28

**Authors:** Ron Shavit, Talma Kushnir, Uri Gottlieb, Shmuel Springer

**Affiliations:** 1The Neuromuscular & Human Performance Laboratory, Faculty of Health Sciences, Department of Physiotherapy, Ariel University, Ariel 40700, Israel; 2Department of Psychology, Ariel University, Ariel 40700, Israel; 3Adelson School of Medicine, Ariel University, Ariel 40700, Israel

**Keywords:** low back pain, clinician self-efficacy, outcome measures, reliability, validity, factor analysis

## Abstract

Background: Clinician self-efficacy may be an important factor in the success of treatment for low back pain (LBP), which has unique clinical features and a high prevalence rate. Therefore, it is important to assess clinicians’ self-efficacy in this particular condition. The Physiotherapist Self-Efficacy (PSE) questionnaire was designed to measure self-efficacy of physiotherapy students. Objectives: To translate and trans-culturally adapt the PSE into Hebrew, to adjust the questionnaire to assess clinicians’ self-efficacy in the treatment of LBP, and to assess the construct validity and reliability of the PSE in the Hebrew version. Methods: After adjustment for LBP and cross-cultural adaptation, test–retest reliability was assessed with 140 physiotherapists. The analyses used included exploratory factor analysis for structural validity, Cronbach’s alpha for internal consistency, and intraclass correlation coefficients (ICC) for test–retest reliability. Results: Factor analysis revealed a unidimensional structure with an acceptable model fit. The PSE translated into Hebrew exhibited a very high internal consistency (α = 0.93) and excellent test–retest reliability (ICC = 0.94). The standard error of measurement (SEM) and minimal detectable change (MDC) were 1.75 and 4.85, respectively. Conclusions: The Hebrew-translated PSE showed adequate validity and excellent reliability, indicating its suitability to measure clinician self-efficacy in treating patients with LBP.

## 1. Introduction

Treatment outcomes in physiotherapy (PT) are influenced by a variety of factors, some of which depend solely on the clinician [1,2]. Modern PT requires the practitioner to make clinical decisions on the basis of a constantly evolving body of knowledge [3]. Self-efficacy, a person’s belief that they are capable of performing a given task, is an important factor linking acquired knowledge to the practical application of skills [4]. People with low self-efficacy about the skills they possess are more likely to avoid professional tasks and are less likely to persist when encountering challenges and impediments [5]. Clinician self-efficacy is considered a predictor of successful performance in many situations [6] and it may serve as an important mediator of successful treatment outcomes [7,8]. Therefore, higher self-efficacy should be one of the ultimate goals of educational programs in PT, and it is important to assess it with validated instruments.

The self-efficacy of physiotherapists (PTs) has been examined in several studies [8,9,10,11,12]. Black et al. [8] measured the self-efficacy of 28 third-year PT students after they completed a 4 h motivational interviewing training. Since self-efficacy is situation specific [13], they used a self-developed questionnaire with three close-ended and two open-ended questions to assess students’ confidence in motivating patients to engage in physical activity. Nithman et al. [9] also used a self-developed questionnaire to examine the impact of intensive care unit simulation training on students’ perceived readiness for clinical placements. The questionnaire contained nine questions rated on a 5-point Likert scale and was developed by an expert panel of four university faculty members. However, in all of these studies [8,9,10,11,12], the self-developed scales were not validated, which significantly limits the ability to conclude and make further use of the proposed measurement tools. 

In contrast to these unvalidated scales, the Physiotherapist Self-Efficacy (PSE) questionnaire (Appendix A) was designed and validated to measure self-efficacy of PT students [14]. The PSE contains 13 items measuring self-efficacy beliefs in three PT clinical domains: musculoskeletal, cardiorespiratory, and neurological. The items of the PSE are rated on a five-point Likert scale indicating the degree of confidence in performing a described task (1 = very little confidence; 5 = a lot of confidence). The methodological quality of the PSE has been examined, and the component structure of the PSE suggests that the self-efficacy of PT students is not general but specific to a clinical domain [14]. Thus, when using the PSE as an outcome measure, the rater must select one of the three clinical domains. The PSE was used by van Lankveld et al. [15] to compare the outcomes of two educational approaches for PT students. They found that self-directed learning and traditional classroom-based learning resulted in equal self-efficacy at the end of the second year. In addition, Campbell et al. [16] conducted a study to investigate the readiness of physiotherapy doctoral students for clinical experiences. The authors concluded that self-efficacy can be used to identify students who need additional supervision during traditional and telemedicine clinical experiences. In contrast to van Lankveld et al. [15], they used a general validated but not domain specific self-efficacy scale [17].

Low back pain (LBP) is the most common complaint in musculoskeletal clinical practice and the leading cause of activity limitations and work absence in most countries of the world [18]. Since 1990, the number of years with disability due to LBP has increased by more than 50%, especially in low- and middle-income countries [18,19]. Since clinician self-efficacy may be an important mediator of successful treatment outcomes [7,8] and LBP has unique clinical features and a high prevalence rate, it could be valuable to assess self-efficacy in this specific clinical domain using the PSE.

Currently, the PSE is not available in Hebrew and there is no other self-efficacy measurement tool available in Hebrew for PT students or clinicians. Nowadays, it is recognized that items of a measurement to be used in another language and/or culture must not only be well translated linguistically but also culturally adapted to maintain content validity across different cultures [20]. Furthermore, the PSE has only been used to assess self-efficacy in PT students, but not in practicing clinicians. It is important to assess PSE fluctuation over time as clinicians gain additional experience and pursue postgraduate training. Resnik et al. [2] have shown that post-graduate training and certification can lead to better clinical outcomes in treating people with LBP [2]. The common explanation is that the additional clinical knowledge and skills are responsible for this change in outcomes. However, it can also be assumed that a change in the PTs’ self-efficacy may have led to improved outcomes. This assumption highlights the need to assess the self-efficacy of practicing clinicians treating people with LBP. 

Therefore, this study aimed to (1) translate and trans-culturally adapt the PSE into Hebrew, (2) adjust the PSE so that it can be used to assess clinicians’ self-efficacy in treating LBP, and (3) assess the construct validity and reliability of the Hebrew version of the PSE.

## 2. Methods

The study design followed the COSMIN checklist for outcome measure instruments [21]. We report our methods and results following the Guidelines For Reporting Reliability and Agreement Studies (GRRAS) [22]. The Ariel University Ethics Committee approved this study (number AU-HEA-SS-20220212). 

### 2.1. The Translation Procedure 

We used the original English version of the PSE for translation and adaptation. The English version of the PSE demonstrated excellent measurement properties (Cronbach’s alpha > 0.90, and test–retest reliability intraclass correlation coefficient (ICC) of 0.80) [14]. Hypothesis testing confirmed the construct validity of the original PSE [23]. Before translation, the investigators provided an adjusted English version of the PSE using the term “low back pain” instead of the original three clinical domains. In this study, we wanted to make the PSE more specific and therefore used the term “low back pain” instead of one of the original general domains to better measure physiotherapists’ self-efficacy in treating this unique and complex clinical entity [24,25]. In addition, the term “low back pain” was chosen as it represents the wide range of patients with different clinical presentations who suffer from low back symptoms. 

The translation employed a five-step procedure according to the guidelines for cross-cultural adaptation of self-report measures [20]. In stage 1, two translators independently translated the PSE from English into Hebrew. The translators were PTs with 3 and 10 years of experience who were fluent in both Hebrew and English. In stage 2, the two translated versions were merged into an agreed version. In stage 3, two different PTs with 7 and 15 years of experience who were fluent in both Hebrew and English performed a back-translation into English. In stage 4, a panel of experts consisting of 12 PTs and two translators discussed the differences in the translations and reached a consensus on the final version of the Hebrew PSE. All concerns and discrepancies in the translation were resolved during the discussion. In the forward translation, the expert panel expressed concern about the accuracy and clarity of the Hebrew translation of the term “patient conditions” used in item 13 (“I feel that I am able to deal with the range of patient conditions which may be seen with a low back pain caseload”). The translators used a term that can be translated back into English as “patient health conditions”, whereas 4 of the 12 experts described the translation as not precise enough and too narrow. The expert panel proposed to use the Hebrew term for “patient health and clinical conditions”. The new resolution was discussed and unanimously approved by the expert panel and the translators. There were no concerns or discrepancies in the back translation. In the fifth and final stage (pilot testing), 39 PTs (age 33.1 ± 5.4 years, mean experience of 7.2 ± 6.9 years, 19 females) were interviewed after completing the translated version of the PSE. They were asked about ambiguous or unclear words and the time it took to complete the questionnaire. Additionally, care was taken to ensure that they understood the meaning of each question. The participants did not indicate any particular problems.

### 2.2. Psychometric Evaluation

To assess the psychometric properties of the translated PSE questionnaires, we conducted an online survey using the Qualtrics website [26]. The translated PSE was completed by participants at two different time points, approximately 14 days apart. This time period is considered long enough to avoid recall bias and short enough to prevent unwanted significant measurement changes [27]. Participants were PTs recruited through social media groups and email lists of the authors. The sample size was estimated according to the COSMIN checklist for measurement error and reliability, which recommends more than 100 participants for very good reliability [21]. The survey was available online in March 2022. The survey landing page included a description of the research and contact information for the principal investigator. To ensure anonymity, the participants used a personal code that was based on the last 4 digits of their national identity number. Participants were informed that they were giving informed consent by pressing “Continue” and that they could stop participation at any time. Multiple entries by the same participant were monitored by tracking the IP addresses. At the beginning of the survey, a yes or no question was asked about the license to practice PT in Israel. If a participant answered “no”, the survey was terminated. 

Hebrew is a gendered language that uses binary pronouns, meaning it assigns gender to verbs, nouns, and adjectives. It has been shown that the masculine generic form of Hebrew is a source of bias, thus increasing the likelihood of false conclusions [28]. Therefore, our online version of the PSE questionnaire was administered in a gender-specific language, according to participants’ selection of gender in the demographic questions. 

### 2.3. Statistical Analysis

The structural validity of the translated PSE was assessed using exploratory factor analysis (EFA with principal component analysis and varimax rotation). The Kaiser–Meyer–Olkin test (KMO index) and Bartlett’s sphericity test were used [29,30] to examine correlations between items and identify coherent factors of the instrument. KMO values of 0.7–0.8 and 0.8–0.9 were interpreted as good and excellent, respectively [29,30,31,32]. Eigenvalue value was set to 1 and above. In addition, the scree plot inflexion point was also examined to determine the number of factors. We used factor loading patterns to identify and extract items. The minimum factor loading criteria were set at 0.30 and above [33]. An item with a commonality below 0.4 with the extracted factors was considered invalid [31]. Internal consistency was calculated using Cronbach’s alpha (α), while the intraclass correlation coefficients (ICC) were used to calculate test–retest reliability. The following scale was used to determine the ICC score: excellent (>0.90), good (>0.75), and moderate (0.50–0.75) [34]. A floor and ceiling effects were assumed if more than 15% of participants scored the lowest or highest on the PSE. The pooled standard deviation formula was used to calculate the standard error of measurement (𝑆𝐸𝑀 = 𝑆𝐷 × √1 − 𝐼𝐶𝐶). Minimal detectable change (MDC) was calculated using the following formula: 𝑀𝐷𝐶 = 𝑆𝐸𝑀 × 1.96 × √2 [34]. Spearman tests were used to assess the correlation between age and experience and PSE score, two independent *t*-tests were used to estimate the effects of gender and postgraduate academic education on PSE score, and a one-way ANOVA was performed to assess the effect of the workplace (5 different categories). Data were analyzed using IBM SPSS Statistics for Windows (version 27.0. Armonk, NY, USA).

## 3. Results

### 3.1. Participants 

Table 1 describes the characteristics of the participants. A total of 314 Hebrew-speaking PTs completed the survey, of whom, 140 completed the survey at the second time point. All questionnaires were valid, and there were no missing data. The age of the participants ranged from 29 to 72, and 55.1% self-identified as female. Experience ranged from less than 1 year to over 42 years, and 92 participants (29%) had postgraduate academic education. 

### 3.2. PSE Hebrew Reliability

The internal consistency obtained with Cronbach’s alpha indicated an excellent value of 0.93. Test–retest reliability of the Hebrew translated PSE was also excellent, ICC = 0.94 (95% CI, 0.91–0.95). The SEM value was 1.75 points, and the MDC was 4.85.

### 3.3. Internal Structure and Construct Validity

The Kaiser–Meyer–Olkin test confirmed the suitability of the sample for analysis with excellent values (KMO = 0.94). The Bartlett’s sphericity test (chi-squared (78) = 2526.933, *p* < 0.001) indicated that the correlations between the items of the translated PSE were sufficient for the analysis. The criterion of extraction of factors with eigenvalues >1 showed the presence of one factor associated with the 13 items of the instrument. The graph of scree plot sedimentation below presents the distribution of the eigenvalues, and the one component that is positioned before the inflection point are presented in Figure 1. This factor explains 56.48% of the total variance of the participants’ responses. This value is satisfactory, as it should explain at least 50% of the total variance of the PSE [35,36]. The EFA results showed that all commonalities were over 0.30, while 10 out of 13 items’ initial commonalities and 8 out of 13 items extracted commonalities exceeded 0.5. Since all items loaded significantly, no items were removed from the analysis. Table 2 describes the EFA principal axis factoring commonalities.

### 3.4. PSE Score Results

The mean PSE score for the entire sample was 51 ± 14. There was a moderate positive significant correlation between age and years of experience and PSE scores (r = 0.30 and 0.36, respectively, *p* < 0.001). No significant gender differences were found. The PSE scores of participants with postgraduate academic education were higher than those of participants with bachelor’s degrees (52.63 and 50.53, respectively, *p* = 0.016). The ANOVA that tested the effect of the workplace showed a significant result (*p* < 0.001). The post hoc analysis demonstrated that PTs working in a public or private outpatient clinic had significantly higher PSE scores (PSE = 51.38 and 52.74, respectively) compared to PTs working in a hospital (PSE = 45.80, *p* < 0.001). PTs working in an inpatient rehabilitation setting had a significantly lower score compared to PTs working in a private outpatient clinic (PSE = 47.32, *p* = 0.021).

## 4. Discussions

This study demonstrates the successful trans-cultural adaptation of the adjusted PSE into Hebrew. No particular difficulties were encountered during the translation process, and the translated questionnaire was found to be highly reliable and valid. Furthermore, in this study, we demonstrated a specific adaptation of the PSE, providing the opportunity to assess clinician self-efficacy in treating LBP, which is the most common complaint in musculoskeletal practice [18].

The obtained Cronbach’s alpha value of the translated PSE showed excellent internal consistency, indicating that the scale was coherent and homogeneous. Although Cronbach’s alpha values were specific to the particular group of responders and cannot be purely generalized [37], the relatively large sample size in this study, which exceeded COSMIN recommendation [21], may further strengthen the reliability of the translated PSE. The EFA of the translated PSE questionnaire resulted in one extracted factor with eigenvalues > 1, explaining 56.48% of the total variance. This unidimensional structure allowed for a single summated score. All items displayed moderate-to-high loadings, except item 11, which still had a loading above 0.3 and therefore was not removed [33]. These results strengthen the structural validity of the PSE [34,35,36].

Since LBP is the most common complaint in musculoskeletal clinical practice and the leading cause of disability and works absence [18], researchers should invest time and effort into better understand this condition and develop effective treatments. To our knowledge, this is the first study to examine clinician self-efficacy in treating LBP. 

The interpretation of PSE values has not been fully clarified. For example, categorical thresholds have not yet been established. Therefore, we have not been able to interpret the level of PSE values and categorize them as low, moderate, or high. In addition, it is not clear to what extent a change in PSE score represents a meaningful change in clinician self-efficacy despite our calculation of MDC. We therefore recommend that categorical thresholds be established in future studies and that the minimum clinically important difference scores of the PSE be calculated in order to make better use of the instrument in studies and educational settings. Our results showed that age and experience were moderately correlated with PSE scores. This may suggest that repeated exposure to patients with LBP increases clinician self-efficacy. Our findings also showed that PTs working in public or private outpatient clinics had significantly higher PSE scores than PTs working in hospitals. In addition, PTs who worked in an inpatient rehabilitation facility had a significantly lower score compared with PTs who worked in a private outpatient clinic. These results are consistent with Bundara’s assertion that self-efficacy is situation specific and not general [13], as LBP cases are almost exclusively treated in outpatient clinics, but in some cases patients with LBP may also be treated in inpatient rehabilitation facilities. 

PTs with postgraduate academic education had higher PSE scores than PTs with bachelor’s degrees. However, the difference was not greater than the MDC [34]. This may further emphasize that better clinician self-efficacy in treating patients with LBP is related to specific exposure to such cases. Therefore, postgraduate academic education does not necessarily lead to a meaningful improvement in self-efficacy without the relevant clinical experience.

Clinician self-efficacy is thought to lead to better treatment outcomes [7,8]. However, to our knowledge, no study has demonstrated this association specifically while using a validated outcome measure for self-efficacy. We therefore recommend that future studies examine whether higher clinician self-efficacy leads to better treatment outcomes. If such an association will be demonstrated, clinician self-efficacy should serve as an outcome measurement for postgraduate clinical education programs as it is easier to apply in contrast to other means. By using PSE as an assessment tool, postgraduate clinical education programs can be evaluated for their effectiveness in improving the future performance of participating clinicians.

As with any study, there are some limitations to note. First, the study consisted exclusively of PTs, and therefore the results can be only generalized to this population and should be taken with caution if applying the PSE to other professions who treat patients with LBP. Second, participants were recruited through social media groups and email lists, which may lead to selection bias and not represent the general Israeli PT population. To overcome this issue, the study sample was larger than the sample recommended by the COSMIN guideline [21]. Finally, in this study, the PSE was adjusted to a specific condition (LBP). This might reduce the applicability of the Hebrew version of the PSE to other clinical conditions. 

## 5. Conclusions

The Hebrew version of the adjusted Physiotherapist Self-Efficacy questionnaire in the treatment of low back pain is valid and reliable. This allows the use of the PSE in the population of interest and its application as an outcome measure in PT continuing education on low back pain treatment.

## Figures and Tables

**Figure 1 healthcare-11-00085-f001:**
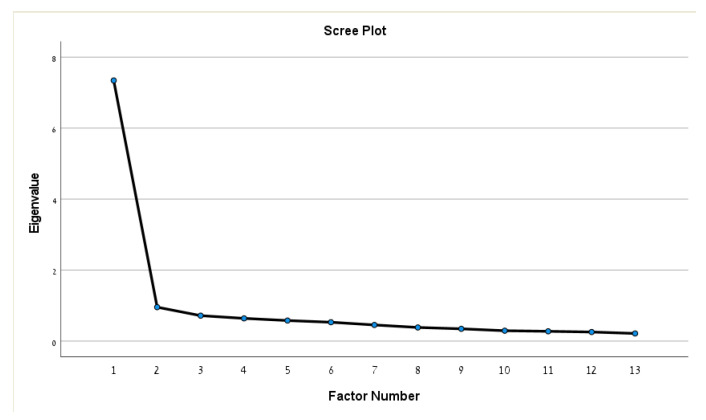
The sedimentation plot (scree plot).

**Table 1 healthcare-11-00085-t001:** Demographics of the survey respondents (first survey completion, N = 314).

Age (Years)	38 ± 9.8
Gender	
Female	173 (55.1%)
Male	141 (44.9%)
Experience (years)	10 ± 9.9
Postgraduate academic education	92 (29%)
Employment	
Health maintenance organizations’ outpatient clinic	169 (53.6%)
Private practice	77 (24.5%)
Hospital setting	20 (6.4%)
Inpatients rehabilitation center	19 (6.2%)
Others (non-specified)	29 (9.3%)

**Table 2 healthcare-11-00085-t002:** EFA principal axis factoring communalities.

PSE Items	Initial	Extraction
1	0.692	0.668
2	0.560	0.532
3	0.516	0.477
4	0.530	0.476
5	0.630	0.573
6	0.637	0.644
7	0.576	0.552
8	0.445	0.388
9	0.650	0.635
10	0.560	0.505
11	0.430	0.347
12	0.662	0.659
13	0.441	0.434

EFA = exploratory factor analysis; PSE = Physiotherapist Self-Efficacy (PSE) questionnaire.

## Data Availability

The data presented in this study are unavailable due to privacy and ethical restrictions.

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
