# Peer review of "Cross-Cultural Adaptation, Reliability, and Validity of a Hebrew Version of the Physiotherapist Self-Efficacy Questionnaire Adjusted to Low Back Pain Treatment"

_healthcare, 2022, doi:10.3390/healthcare11010085_

Round 1

Reviewer 1 Report

The manuscript shows the “Cross-cultural adaptation, reliability, and validity of a Hebrew version of the Physiotherapist Self-Efficacy questionnaire adjusted to Low Back Pain treatment”. It is an interesting topic but I have some suggestions for you.

Line 42: add a full stop after the Word “al”

Line 46,73: idem

I do not understand why do you decide to change the three clinical domains into “low back pain”. Is it too general? I assume that you want to make the PSG more specific but could you are wasting information from the original questionnaire in that way? One option could it be to validate the original one and after that to adapt it to lo w back pain.

Regarding table 1: the “experience” range from 10 ± 9.9. May I assume that there is some participants with 0.1 year of experience? Could this data affect the results?

Were all questionnaires valid? Is there any missing data? If the answer is yes, please state that in your manuscript.

Line 181: add and space between Figure and 1.

Regarding the point 3.4. I think it should be interesting that you mention more in detail the different places were these physiotherapists work.

There is a few mention about low back pain into discussion section. Again I wonder why did you decide to make this change in the original version. You should add more references that support your decision.

I would like to see a copy of the original version of this scale, maybe as an annex, I think it could help the reader.

There are no author contributions, funding… at the end of the manuscript.

Please review MDPI rules for references.

Reviewer 2 Report

First of all, congratulations for the research work carried out, I will now mention a series of changes and recommendations with the aim of obtaining clearer and more precise information on the results.

The work presents a high quality of elaboration in terms of methodology, contents and results. Regarding the introduction, it would be convenient to increase the number of references and the length of the introduction, introducing also the background paragraph with other tests commonly used in treatment self-efficacy scales, in order to provide a broader view of the existing scientific literature. also through a quick review of it.

From a methodological point of view, I recommend specifying if the low back pain evaluation is for the most common condition "non-specific low back pain" or for other conditions such as disc herniation, post-surgery ..... Or if it can be used in any situation as long as the patient has low back pain.

The results are well represented both graphically and from a descriptive point of view, however, it is recommended to develop the Others section of table 1 in the text.

The discussion is well structured, but I would suggest: 1) to increase the references because they are too few compared to the reported contents; 2) to make known what future research developments can be.

The conclusions are well readapted according to the results shown in the study. Perhaps I would also add in a more specific way for which type of patients this adjustment is indicated. The term low back pain can be considered unspecific.
